# Characterization of human translesion DNA synthesis across a UV-induced DNA lesion

Mark Hedglin[†], Binod Pandey[†], Stephen J Benkovic*

Department of Chemistry, The Pennsylvania State University, University Park, United States

**Abstract** Translesion DNA synthesis (TLS) during S-phase uses specialized TLS DNA polymerases to replicate a DNA lesion, allowing stringent DNA synthesis to resume beyond the offending damage. Human TLS involves the conjugation of ubiquitin to PCNA clamps encircling damaged DNA and the role of this post-translational modification is under scrutiny. A widely-accepted model purports that ubiquitinated PCNA recruits TLS polymerases such as pol η to sites of DNA damage where they may also displace a blocked replicative polymerase. We provide extensive quantitative evidence that the binding of pol η to PCNA and the ensuing TLS are both independent of PCNA ubiquitination. Rather, the unique properties of pols η and δ are attuned to promote an efficient and passive exchange of polymerases during TLS on the lagging strand.

## Introduction

In eukaryotes, the replicative DNA polymerases (pols), ε and δ, are responsible for replicating the leading and lagging strands, respectively, and anchor to PCNA sliding clamps encircling the DNA to achieve the high degree of processivity required for efficient DNA replication. This association, referred to as a holoenzyme, tethers the pol to DNA, substantially increasing the extent of continuous replication (*Hedglin et al., 2013a*). However, the stringent replicative pols cannot accommodate distortions to the native DNA sequence such as such as cyclobutane pyrimidine dimers (CPDs), the major DNA lesions resulting from ultraviolet (UV) irradiation. Upon encountering these lesions, DNA synthesis on the afflicted template abruptly stops but the replication fork progresses onward, exposing the damaged template. Fork progression eventually stalls and replication protein A (RPA) coats the exposed ssDNA, protecting it from cellular nucleases (*Hedglin and Benkovic, 2015*).

Such arrests may be overcome by translesion DNA synthesis (TLS) where the replicative pol is exchanged for a TLS pol that binds to the resident PCNA and replicates the damaged DNA (*Sale et al., 2012*). With a more open pol active site and the lack of proofreading activity, TLS pols are able to support stable, yet potentially erroneous, nucleotide incorporation opposite damaged templates allowing DNA synthesis by the replicative pol to resume (*Lange et al., 2011*). In humans, TLS involves the conjugation of single ubiquitin moieties (that is monoubiquitination) to PCNA clamps encircling blocked primer/template (P/T) junctions and at least seven TLS pols with varying fidelities. However, remarkably low error rates are observed in vivo after exposure to various DNA-damaging agents, indicating a highly efficient process (*Hedglin and Benkovic, 2015*; *Lange et al., 2011*; *Yoon et al., 2012*). Currently, the mechanism by which polymerase exchange occurs during human TLS and the role of monoubiquitinated PCNA in this process are unknown.

Monoubiquitination of PCNA is essential for optimal TLS activity in mammalian cells (*Hendel et al., 2011*). The signal for this post-translational modification (PTM) is the buildup and persistence of RPA-coated ssDNA during S-phase. Such structures recruit the Rad6/Rad18 complex

*For correspondence: sjb1@psu.edu

[†]These authors contributed equally to this work

Competing interests: The authors declare that no competing interests exist.

to the exposed DNA template where it catalyzes monoubiquitination of lysine residue(s) K164 of the PCNA ring encircling the blocked P/T junction upstream. Hence, this PTM is a generic response that may be elicited by any agent that uncouples DNA synthesis by the replicative pols from replication fork progression, even those that do not modify the DNA at all (*Hedglin and Benkovic, 2015*). Most studies on the role of monoubiquitinated PCNA during TLS have focused on members of the Y-family of TLS pols, pol η in particular. In human cells, pol η is responsible for the error-free replication of CPDs (*Yoon et al., 2009*). The magnitude of this feat is underscored by xeroderma pigmentosum variant (XPV), a human autosomal recessive genetic disorder in which the *xpv* gene encoding polη is either mutated or deleted, leading to extreme UV sensitivity and skin cancer predisposition (*Masutani et al., 1999*). Following UV irradiation, PCNA encircling damaged DNA is monoubiquitinated during S-phase and pol η co-localizes with PCNA in replication factories (foci) on damaged DNA. Both activities are imperative for TLS following UV irradiation. All Y-family pols contain one or more PCNA-binding domains and at least one ubiquitin-binding domain (UBD) (*Sale et al., 2012*). The seminal in vivo studies suggested the widely-accepted 'recruitment/displacement' model that the ubiquitin moieties attached to PCNA serve to directly recruit pol η (via its UBD) to sites of DNA damage where it may also displace a blocked replicative pol (*Bienko et al., 2005*; *Kannouche et al., 2004*). However, one cannot extrapolate these findings to conclude unequivocally that a direct interaction between pol η and a ubiquitin conjugated to PCNA is required or even occurs during human TLS (*Sabbioneda et al., 2009*). In fact, the collective in vivo evidence suggests otherwise. Firstly, pol η co-localizes with PCNA in replication foci during unperturbed S-phase in human cells and this activity requires the UBD of pol η even though PCNA monoubiquitination is absent (*Bienko et al., 2005*). Secondly, pol η accumulates into replication foci after UV irradiation independently of PCNA monoubiquitination (*Göhler et al., 2011*; *Sabbioneda et al., 2008*) and does not require the UBD of pol η (*Despras et al., 2012*). Rather, monoubiquitinated PCNA and the UBD of pol η independently retain pol η within replication foci, increasing the residence time of localized pol η. Other reports arrived at the same conclusion but challenged the requirement of the UBD of pol η for TLS following UV irradiation (*Acharya et al., 2008*, *2010*). Thirdly, in vivo studies on various eukaryotes irradiated with UV suggest that the requirement for PCNA monoubiquitination depends on the location of TLS relative to the replication fork; TLS at/near a stalled replication fork is independent of PCNA monoubiquitination (*Edmunds et al., 2008*; *Temviriyanukul et al., 2012*) while TLS behind a re-started and progressing replication fork requires PCNA monoubiquitination (*Daigaku et al., 2010*; *Karras and Jentsch, 2010*). How then does polymerase switching and TLS occur and what is the role of PCNA monoubiquitination in this process?

We reasoned that TLS pols must bind tighter to monoubiquitinated PCNA than to native PCNA in order for the recruitment/displacement model to be operative. In this report, we performed extensive quantitative studies on human pols δ and η to delineate TLS on the lagging strand. The results clearly demonstrate that the binding of pol η to PCNA and DNA synthesis by a pol η holoenzyme are both independent of PCNA monoubiquitination, refuting the recruitment/displacement model for human TLS. Furthermore, these studies reveal that the unique properties of pols η and δ are attuned to promote a passive and efficient exchange of pols that is independent of PCNA monoubiquitination. Altogether, these studies reveal a novel mechanism for human TLS and direct future studies on the role of PCNA monoubiquitination.

## Results

### Pol η binds to PCNA and monoubiquitinated PCNA with equivalent affinities

Human pol η (referred to herein as pol η) is a single subunit comprised of an N-terminal (residues 1–475) 'catalytic core' and a C-terminal portion (residues 476–713) containing all protein-protein interaction motifs; a UBD and three PCNA-binding domains that each resemble those from pol δ (*Figure 1—figure supplement 1*). We utilized FRET to characterize the interaction between pol η and PCNA. FRET between Cy3-pol η and Cy5-PCNA is only observed in the presence of both proteins (*Figure 1—figure supplement 2*) and this signal was utilized to determine the affinity of the pol η•PCNA interaction (*Figure 1A*, left panel). As observed in *Figure 1B*, Cy3-pol η binds to Cy5-PCNA with a $K_D$ = 120 ± 9.06 nM (*Table 1*). To directly compare the binding affinities of pol η for

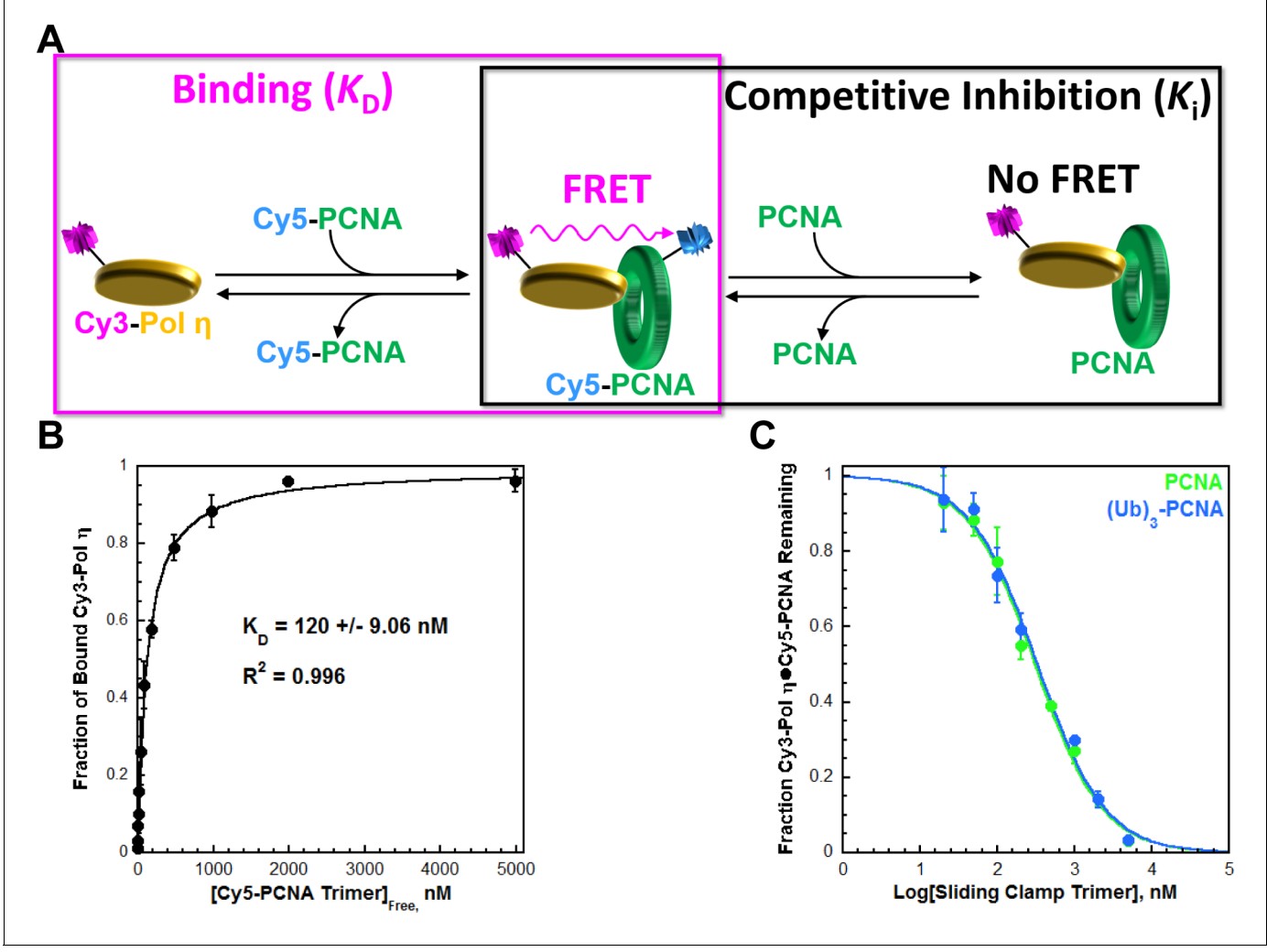

**Figure 1.** Characterizing the interaction between pol η and PCNA. Human pol η contains a UBD and three PCNA-binding domains that each resemble those from pol δ (**Figure 1—figure supplement 1**). We utilized FRET (**Figure 1—figure supplement 2**) to characterize the interaction between full-length, human pol η and PCNA. (**A**) Schematic representation of the equilibrium binding assay (left panel) to measure the binding affinity of Cy3-pol η for Cy5-PCNA (**Figure 1B**) and the competitive inhibition assay (right panel) to measure the binding affinity of an unlabeled PCNA for Cy3-pol η (**Figure 1C**). (**B**) Cy3-pol η was kept constant, titrated with Cy5-PCNA, and the fraction of bound Cy3-Pol η was measured. Data was plotted as a function of free Cy5-PCNA concentration and each point represents the average ± SD of 3 independent experiments. Fitting to a one-site binding model, yields a $K_D$ of 120 ± 9.06 nM. (**C**) The Cy3-Pol η•Cy5-PCNA complex was pre-assembled, titrated with an unlabeled PCNA, and the fraction of the Cy3-Pol η•Cy5-PCNA complex remaining was measured. Data was plotted versus the log concentration (in nM) of the respective unlabeled PCNA and each point represents the average ± SD of at least 3 independent experiments. $IC_{50}$ values for PCNA (green) and $(Ub)_3$-PCNA (blue) were obtained by fitting to a dose-response inhibition model. As a control, this assay was repeated with pol η (**Figure 1—figure supplement 3**). $K_i$ values were calculated from the $IC_{50}$ for each competitor and reported in **Table 1**. $(Ub)_3$-PCNA contains a single ubiquitin moiety on K164 of each monomer within a homotrimeric clamp ring (**Figure 1—figure supplement 4**).

The following figure supplements are available for figure 1:

**Figure supplement 1.** PCNA-binding domains of human pol δ and η.

**Figure supplement 2.** Monitoring the interaction between Pol η and PCNA through FRET.

**Figure supplement 3.** Cy3 does not compromise the affinity of pol η for PCNA.

**Figure supplement 4.** Monoubiquitinated PCNA contains a single ubiquitin moiety on K164 of each monomer within a homotrimeric clamp ring.

**Table 1.** Binding affinities ($K_D$ or $K_i$) calculated from equilibrium assays carried out in the present study.

| Binding assays | Substrate | Ligand | $K_D$, nM | Figure[*] |
|---|---|---|---|---|
| | Cy3-Pol η | Cy5-PCNA | 120 ± 9.06 | 1B |
| | FLUOR-P/T DNA | Pol η | 29.7 ± 4.26 | 2 |
| | | Pol δ | ~537 | 2FS2 |
| | Forked Cy3-P/T DNA•Cy5-PCNA | Pol η | 28.0 ± 5.38 | 3FS1 |
| | | Pol δ | < 10.0 | 3FS1 |
| Competitive inhibition assays | Complex | Competitor | $K_i$, nM | Figure[*] |
| | Cy3-Pol η•Cy5-PCNA | PCNA | 111 ± 7.78 | 1C |
| | | (Ub)$_3$-PCNA | 119 ± 10.5 | 1C |
| | | Pol η | 112 ± 15.5 | 1FS3 |
| | P/T DNA•PCNA•Pol δ | 'Dead' pol η | 31.7 ± 5.12 | 3 |
| | P/T DNA•(Ub)$_3$-PCNA•Pol δ | 'Dead' pol η | 33.5 ± 5.04 | 3 |

[*]*Figure 2—figure supplement 2*, *Figure 3—figure supplement 1*, and *Figure 1—figure supplement 3* are abbreviated as 2FS2, 3FS1, and 1FS3, respectively.

PCNA and monoubiquitinated PCNA, we carried out a FRET-based competition experiment (*Figure 1A*, right panel) by titrating a pre-assembled Cy3-Pol η•Cy5-PCNA complex with an unlabeled PCNA competitor. As observed in *Figure 1C*, the fraction of Cy3-Pol η•Cy5-PCNA complex decreased with PCNA concentration, indicating that PCNA competes with Cy5-PCNA for binding to Cy3-Pol η (IC$_{50}$ = 308 ± 19.8 nM). A $K_i$ of 111 ± 7.78 nM is calculated from the IC$_{50}$ for PCNA. As a control, this assay was repeated with pol η (*Figure 1—figure supplement 3*), yielding a $K_i$ of 112 ± 15.5 nM. These $K_i$ values are identical and both in excellent agreement with the $K_D$ for the Cy5-PCNA•Cy3-Pol η interaction (*Table 1*). Together, this validates the experimental approach. Monoubiquitinated PCNA (referred to herein as (Ub)$_3$-PCNA) contains a single ubiquitin moiety on K164 of each monomer within a homotrimeric clamp ring (*Figure 1—figure supplement 4*). When (Ub)$_3$-PCNA was utilized as the unlabeled competitor, a $K_i$ of 119 ± 10.5 nM was obtained, in excellent agreement with the $K_i$ measured for PCNA (*Table 1*). Thus, pol η binds to PCNA and (Ub)$_3$-PCNA with equivalent affinities.

### The DNA binding affinity of pol η drives polymerase exchange

Next, the interaction between pol η and P/T DNA was characterized by fluorescence anisotropy. The fractional saturation of fluorescein-labeled P/T DNA (FLUOR-P/T DNA, *Figure 2—figure supplement 1*) increased with pol η concentration (*Figure 2*) and displayed hyperbolic behavior with a $K_D$ = 29.7 ± 4.26 nM. This value is similar to the $K_D$ reported for the catalytic core of human pol η (38 ± 4 nM), suggesting that the C-terminus does not contribute significantly to the DNA binding affinity (*Washington et al., 2003*). Furthermore, the $K_D$ for P/T DNA indicates that pol η binds ~4 fold tighter to P/T DNA than to a PCNA (*Table 1*). This behavior contrasts that observed for pol δ, which will not form a stable complex with P/T DNA in the absence of PCNA (*Hedglin et al., 2016*). Indeed, when these assays were repeated with pol δ, a significant change in the fractional saturation of the FLUOR-P/T DNA was not observed until a 50-fold excess of pol δ was added (*Figure 2—figure supplement 2*). From the partial binding curve, a $K_D$ of 537 nM is estimated, indicating that pol δ binds much weaker (~18.1 fold) to P/T DNA than pol η does (*Table 1*).

The aforementioned studies reveal that pol η binds to free PCNA independently of PCNA monoubiquitination but much weaker than to P/T DNA. These findings suggest that pol exchange during TLS may be driven by the P/T DNA binding affinity of pol η. To gauge this possibility, we first analyzed formation of the pol η•PCNA•P/T DNA complex (that is a holoenzyme) (*Figure 3—figure supplement 1A*). Pol η holoenzyme formation increased with pol η concentration (*Figure 3—figure supplement 1B*) and displayed hyperbolic behavior with a $K_D$ (28.0 ± 5.38 nM) identical to that measured for pol η binding to P/T DNA (*Table 1*). This suggests that pol η holoenzyme assembly is governed by the affinity of pol η for the P/T DNA, in stark contrast to that observed previously for pol δ

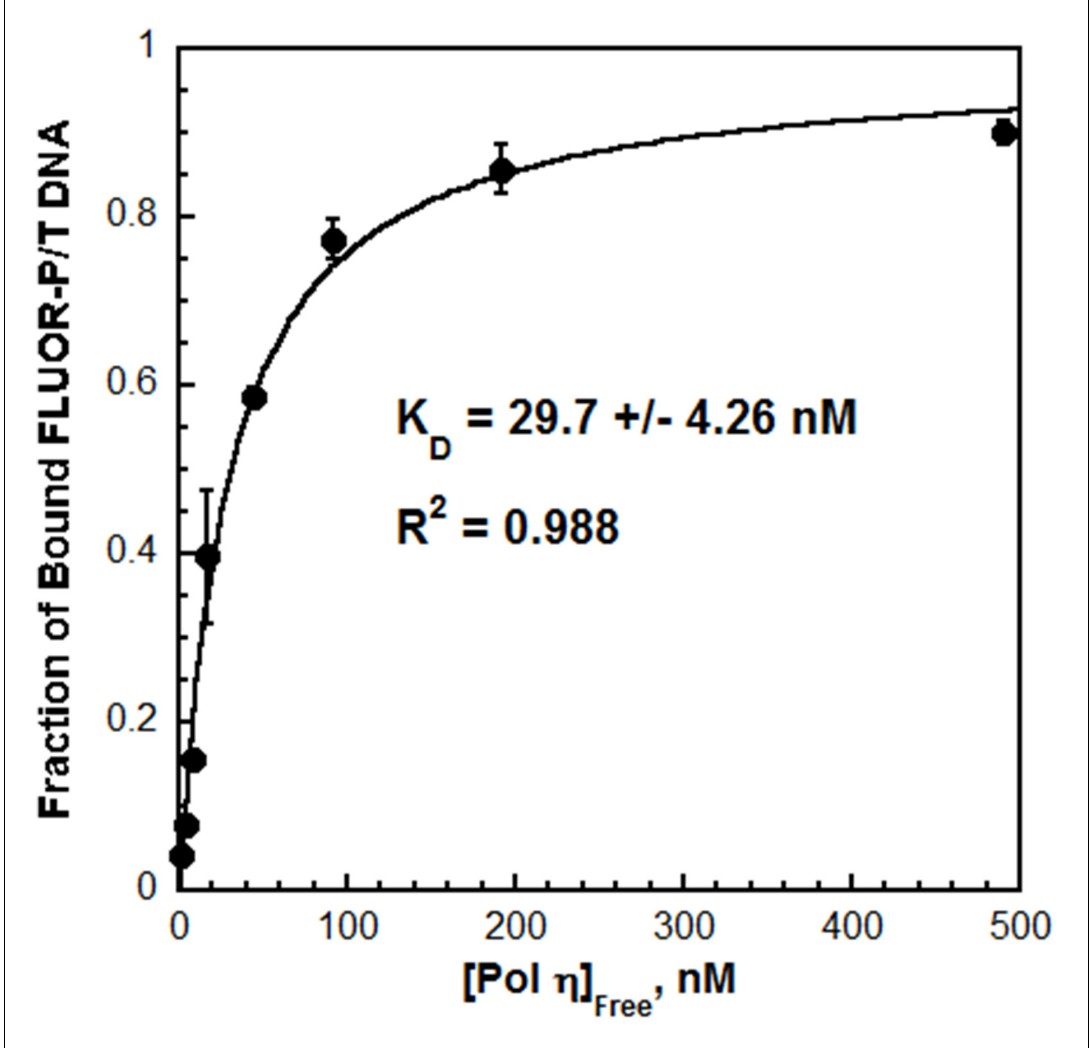

**Figure 2.** The affinity of Pol η for P/T DNA. Fluorescein-labeled P/T DNA (FLUOR-P/T, *Figure 2—figure supplement 1*) was titrated with pol η and the fraction of bound DNA was measured by fluorescence anisotropy. Data was plotted versus the concentration free pol η and each point represents the average ± SD of at least 3 independent experiments. Fitting to a one-site binding model yields a $K_D$ value of 29.7 ± 4.26 nM (*Table 1*). This value is more than 18.1-fold less than the $K_D$ estimated (~537 nM) for pol δ on the same DNA substrate (*Figure 2—figure supplement 2*).

The following figure supplements are available for figure 2:

**Figure supplement 1.** DNA substrates.

**Figure supplement 2.** Pol δ binds much weaker to P/T DNA compared to pol η.

under similar conditions (*Hedglin et al., 2013b*). Pol δ binds to P/T DNA with dramatically low affinity (*Table 1*) yet holoenzyme formation increased linearly, plateaued when the concentrations of pol δ and PCNA were equivalent, and flat-lined thereafter (*Figure 3—figure supplement 1C*). Such behavior indicates that the concentration of the P/T DNA•PCNA complex (100 nM) is much greater than (≥10 fold) the $K_D$ of pol δ for the complex (*Goodrich and Kugel, 2007*). This sets an upper limit of 10 nM for the $K_D$ of pol δ for PCNA encircling a P/T junction, in excellent agreement with the value (7.1 ± 1.0 nM) reported in an independent study (*Zhou et al., 2012*). Thus, pol δ binds much tighter to PCNA encircling a P/T junction than to P/T DNA alone (*Table 1*). Altogether, these studies indicate that assembly of a pol δ holoenzyme is governed by the tighter affinity of pol δ for PCNA while pol η holoenzyme assembly is dictated by the tighter affinity of pol η for P/T DNA. Thus, pol

exchange during TLS may entail a competition between pol η binding to the blocked P/T junction and pol δ binding to the PCNA encircling the blocked P/T junction. To test this, we directly monitored pol exchange.

A pol δ holoenzyme was pre-assembled on a P/T DNA substrate (TT P/T, *Figure 3A*) and then titrated with catalytically-inactive ('dead') pol η (*Figure 3B*). This mutant retains all activities except

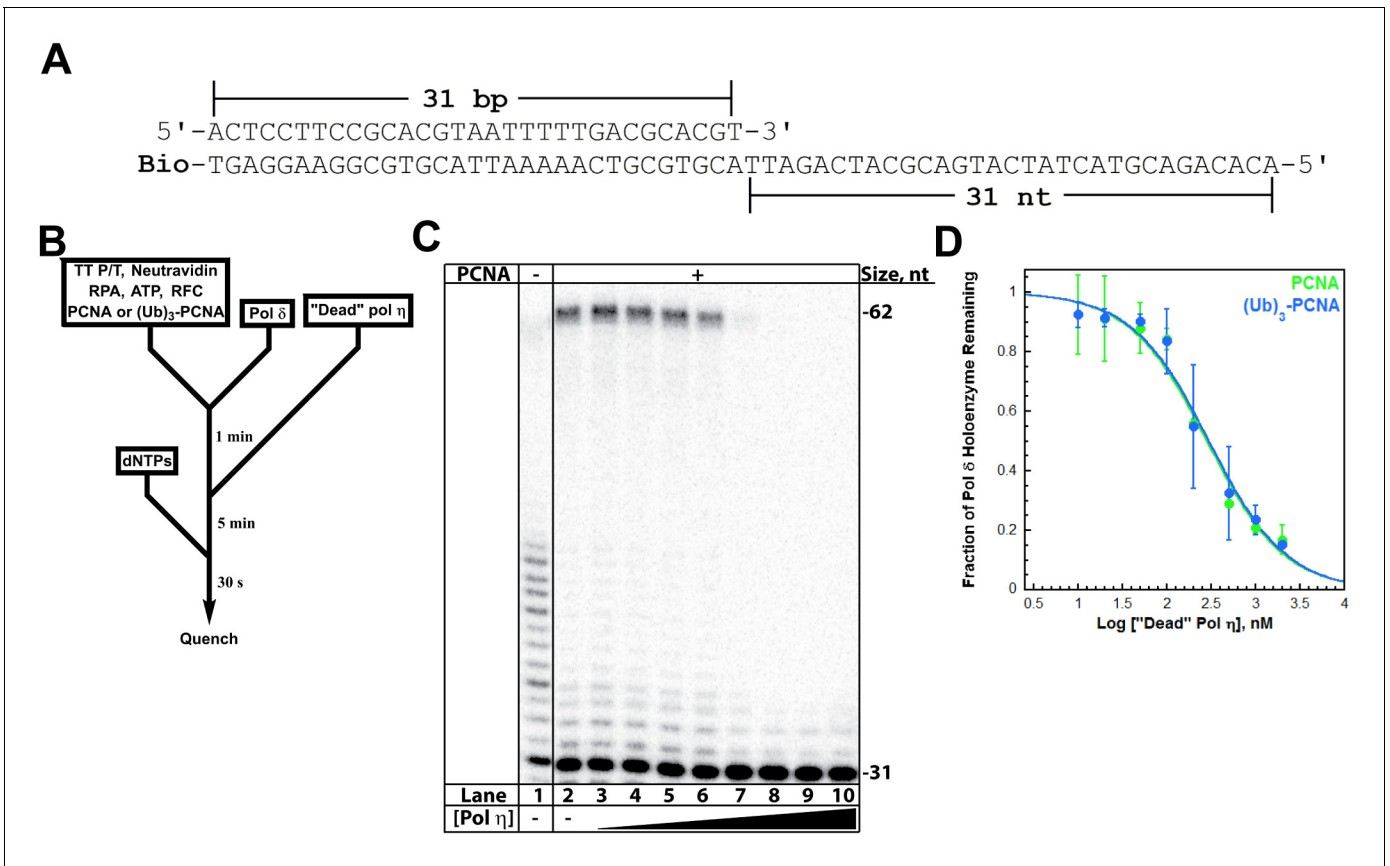

**Figure 3.** Polymerase exchange at a blocked P/T junction. (A) Sequence of the TT P/T DNA substrate that mimics a nascent P/T junction on the lagging strand. The size of the double-stranded DNA region agrees with the size of an initiating P/T and the requirements for assembly of a pol δ holoenzyme. The biotin tag was pre-bound to Neutravidin, preventing PCNA from sliding off the 5′ end of the primer. The ssDNA is consistent with the footprint of a single RPA molecule and was pre-bound with excess RPA. The primer is labeled at the 5′ terminus with [32]P (*Hedglin et al., 2016*). (B) Schematic representation of the experiment to monitor pol exchange. The experiment for pol δ alone was performed identically except for the omission of PCNA, RFC, and 'dead' pol η. (C) 12% denaturing sequencing gel of the primer extension products. The size of the substrate and full-length product are indicated on the right. The full-length product is only generated in the presence of a PCNA (compare lane 1 to lanes 2–10). Thus, production of the full-length product indicates the fraction of pol δ holoenzymes that withstand the influx of 'dead' pol η and, hence, the extent of pol exchange. (D) The fraction of pol δ holoenzyme remaining was plotted as a function of the log concentration of 'dead' pol η. The data for pol δ holoenzymes assembled with either PCNA (green) or (Ub)$_3$-PCNA (blue) is shown and each point represents the average ± SD of 3 independent experiments. IC$_{50}$ values were obtained by fitting the data to a dose-response inhibition model and $K_i$ values (*Table 1*) were calculated from the IC$_{50}$ values. $K_i$ values of 31.7 ± 5.12 nM and 33.5 ± 5.04 nM (*Table 1*) were calculated for pol δ holoenzymes assembled with PCNA and (Ub)$_3$-PCNA, respectively. These equivalent values exactly agree with the $K_D$ for pol η binding to P/T DNA (*Figure 2*) and the $K_D$ for pol η holoenzyme formation (*Figure 3—figure supplement 1*), suggesting that pol η binding to the P/T junction is competing with pol δ binding to PCNA encircling the P/T junction. Repeating this assay with Cy5-PCNA and Cy3-labeled TT P/T DNA and monitoring FRET instead of DNA synthesis (*Figure 3—figure supplement 2*) verified that PCNA does not slide off the unblocked end of the DNA substrate template during the competition.

The following figure supplements are available for figure 3:

**Figure supplement 1.** Differential binding activities drive assembly of the pol η and pol δ holoenzymes.

**Figure supplement 2.** The resident PCNA remains at the blocked P/T junction during the exchange of pol δ for pol η.

DNA synthesis and will compete with pol δ for binding to a P/T junction (*Hedglin et al., 2016*). Indeed, the fraction of pol δ holoenzymes decreased with 'dead' pol η concentration (*Figure 3C*), indicating that this mutant protein competes and exchanges with pol δ for binding to the P/T DNA and/or resident PCNA. From the dose-response curves (*Figure 3D*), $K_i$ values of 31.7 ± 5.12 nM and 33.5 ± 5.04 nM (*Table 1*) are calculated for pol δ holoenzymes assembled with PCNA and (Ub)$_3$-PCNA, respectively. These equivalent values exactly agree with the $K_D$ for pol η binding to P/T DNA and the $K_D$ for pol η holoenzyme formation (*Table 1*), suggesting that pol η binding to the P/T junction is competing with pol δ binding to PCNA encircling the P/T junction. To verify that PCNA remains on the DNA during pol exchange, we repeated these assays with Cy5-PCNA and Cy3-labeled TT P/T DNA and monitored FRET instead of DNA synthesis (*Figure 3—figure supplement 2A*). Despite the exchange of pols, the FRET signal remained constant (*Figure 3—figure supplement 2B*), indicating that PCNA does not slide off the unblocked end of the template during the competition. Altogether, these studies confirm that the exchange of pol δ for pol η at a P/T junction is independent of PCNA monoubiquitination (*Hedglin et al., 2016*) and reveal that the binding affinity of pol η for P/T DNA drives pol exchange during TLS. Next, we probed the effect of PCNA on DNA synthesis by pols δ and η.

## PCNA increases the processivity of pols δ and η

We utilized the TT P/T DNA substrate (*Figure 3A*) to monitor primer extension during a single DNA-binding event (*Figure 4B*), as previously described (*Hedglin et al., 2016*). The first two nucleotides of the template (directly abutting the P/T junction) are both thymine (T) and primer extension is indicated by insertion of a dNTP across from the 3' T of the template. Pol η alone has substantial affinity for P/T DNA (*Table 1*) and will extend the primer (*Figure 4*, panels **C** and **D**). However, extension beyond the 3' T is severely limited such that pol η did not extend the primer more than 6 dNTPs. Thus, pol η alone is inefficient at continuous DNA synthesis. In the presence of PCNA, primer extension is stimulated 2.02 ± 0.173 fold (*Figure 4D*). Identical results were obtained with (Ub)$_3$-PCNA, indicating that the binding of pol η to a PCNA encircling a native P/T junction is independent of PCNA monoubiquitination. Furthermore, primer extension beyond the 3' T is more prominent in the presence of a PCNA such that pol η extends the primer up to 10 dNTPs. Under the conditions of the assay, this behavior can be quantitatively analyzed at single nucleotide resolution (*Hedglin et al., 2016*). The probability of insertion, $P_i$, can be directly measured for each dNTP insertion step, $i$, beyond $i = 1$. For pol η alone, $P_i$ values are quite low (*Figure 5A*). Notably, $P_2$ is only 0.215 ± 0.0333, indicating that the population of pol η that inserts a dNTP across from the 5' T ($i = 2$) prior to dissociation is scarce. This population, $P_2$, is referred to herein as the pol η TLS complex. In the presence of a PCNA, $P_i$ values up to $i = 6$ are increased compared to pol η alone and the observed stimulations are equivalent for PCNA and (Ub)$_3$-PCNA. Notably, the abundance of the pol η TLS complex is increased 2.53 ± 0.425 and 2.75 ± 0.368 fold in the presence of PCNA and (Ub)$_3$-PCNA, respectively. Insertion of dNTPs beyond $i = 6$ is only observed in the presence of a PCNA and the measured $P_i$ values are equivalent in the presence of PCNA and (Ub)$_3$-PCNA. Thus, binding to PCNA increases the $P_i$ for pol η independently of PCNA monoubiquitination but the effect is transient such that the extent of continuous DNA synthesis is only marginally enhanced. The latter behavior is in stark contrast to that observed for pol δ on the same substrate. Pol δ alone binds to P/T DNA with dramatically low affinity (*Table 1*) and will not extend the primer in the absence of PCNA (*Figure 4—figure supplement 1A*), in agreement with previous observations (*Hedglin et al., 2016*). In the presence of PCNA, $P_i$ is stimulated, plateaus at ~1.0 soon after DNA synthesis initiates, and remains constant until the pol δ holoenzyme approaches the end of the DNA template (*Figure 4—figure supplement 1B*). Thus, anchoring to PCNA dramatically enhances the $P_i$ for pol δ from 0.00 to ~1.0 and the effect is perpetual such that the extent of continuous DNA synthesis is substantially increased. Based on the $P_i$ value within the plateau, a pol δ holoenzyme that initiates DNA synthesis from a P/T junction can extend the primer up to 4600 dNTPs prior to dissociation ($0.999^{4600} = 0.010$), in agreement with a previous report (*Hedglin et al., 2016*).

The unique behaviors of pol δ and pol η holoenzymes are quite befitting of their respective cellular functions and suggest distinct mechanisms for the PCNA-dependent stimulation of processivity. Kinetically, $P_i = k_{pol}/(k_{pol} + k_{off})$ where $k_{off}$ and $k_{pol}$ are the rate constants for dissociation of a pol into solution and dNTP insertion, respectively. Pol δ, a replicative B-family pol, inserts complementary dNTPs very fast ($k_{pol} \sim 100$ s$^{-1}$) and maintains very high fidelity (*Hedglin et al., 2016*;

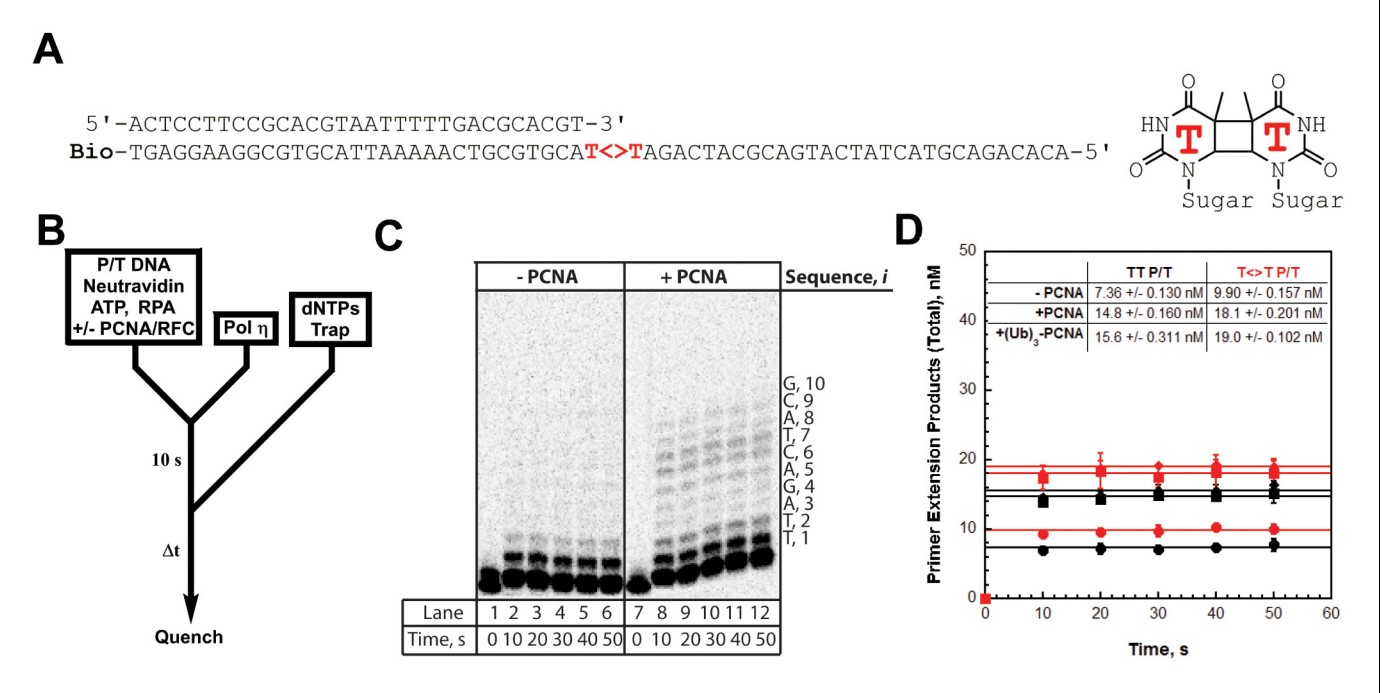

**Figure 4.** DNA synthesis by pol η. (A) Sequence of the T<>T P/T DNA substrate. This substrate is identical to the undamaged DNA substrate (*Figure 3A*) except the native TT sequence has been replaced with a TT-CPD (denoted T<>T, shown on right). (B) Schematic representation of the experiment performed to monitor primer extension by pol η under single turnover conditions. The experiment for pol η alone was performed identically except for the omission of PCNA and RFC. (C) 16% denaturing sequencing gel of the primer extension products for pol η alone (Lanes 1–6) and a pol η holoenzyme assembled with PCNA (lanes 7–12). The sequence of the template for each dNTP insertion step (*i*) is indicated on the right up to *i* = 10. (D) Quantification of the primer extension products observed in the absence of a PCNA (•), in the presence of PCNA (■), and in the presence of (Ub)$_3$-PCNA (♦). Results for undamaged (TT P/T) and damaged (T<>T P/T) DNA substrates are shown. The data is plotted versus time and each data point represents the average ± SD of 3 independent experiments. Data points after t = 10 s were fit to a flat line where the y-intercept reflects the amplitude. Values are reported in the inset. In contrast to that observed for pol η on the undamaged (TT P/T) DNA substrate, DNA synthesis by pol δ was only observed in the presence of PCNA (*Figure 4—figure supplement 1*).

The following figure supplement is available for figure 4:

**Figure supplement 1.** Processive DNA synthesis by pol δ.

---

*Schmitt et al., 2009*). However, pol δ alone has very low affinity for P/T DNA and must anchor to PCNA to efficiently replicate DNA (*Hedglin et al., 2016*). Pol δ binds much tighter to PCNA encircling DNA ($K_D$ <10 nM, *Figure 3—figure supplement 1*) than to DNA alone (~537 nM, *Figure 2—figure supplement 2*) and effectively captures a PCNA ring encircling a nascent P/T junction to initiate DNA synthesis (*Hedglin et al., 2013b*). Thus, anchoring to PCNA dramatically decreases $k_{off}$ such that the $P_i$ of pol δ increases from 0.0 to ~1.0 and the extent continuous dNTP insertion increases by kilobases (*Figure 4—figure supplement 1*). Such behavior is befitting as pol δ is responsible for replicating the lagging strand during S-phase. On the other hand, Pol η, a Y-family TLS pol, also inserts dNTPs very fast (67 s$^{-1}$) but has a compromised fidelity, inserting incorrect dNTPs up to 10$^3$-fold more frequently than pol δ does on undamaged DNA (*Matsuda et al., 2000*; *Washington et al., 2003*). Hence, DNA synthesis by pol η must be restricted to limit replication errors. Pol η has significant binding affinity for P/T DNA (29.7 nM, *Figure 2*) such that it can replicate DNA in the absence of PCNA, albeit with distributive behavior (*Figure 4*). In contrast to pol δ, pol η binds tighter to P/T DNA (29.7 nM, *Figure 2*) that to PCNA (111 nM, *Figure 1*). Thus, anchoring to PCNA marginally decreases $k_{off}$ for pol η such that the $P_i$ increases only ~ 1.4–2.5-fold and the extent of continuous dNTP insertion increases by 4 nucleotides (*Figure 4*). This behavior may be critical for pol η's cellular role in replicating small patches of undamaged DNA during common fragile site replication and somatic hypermutation (*Krijger et al., 2011*; *Rey et al., 2009*). In human cells, pol η is

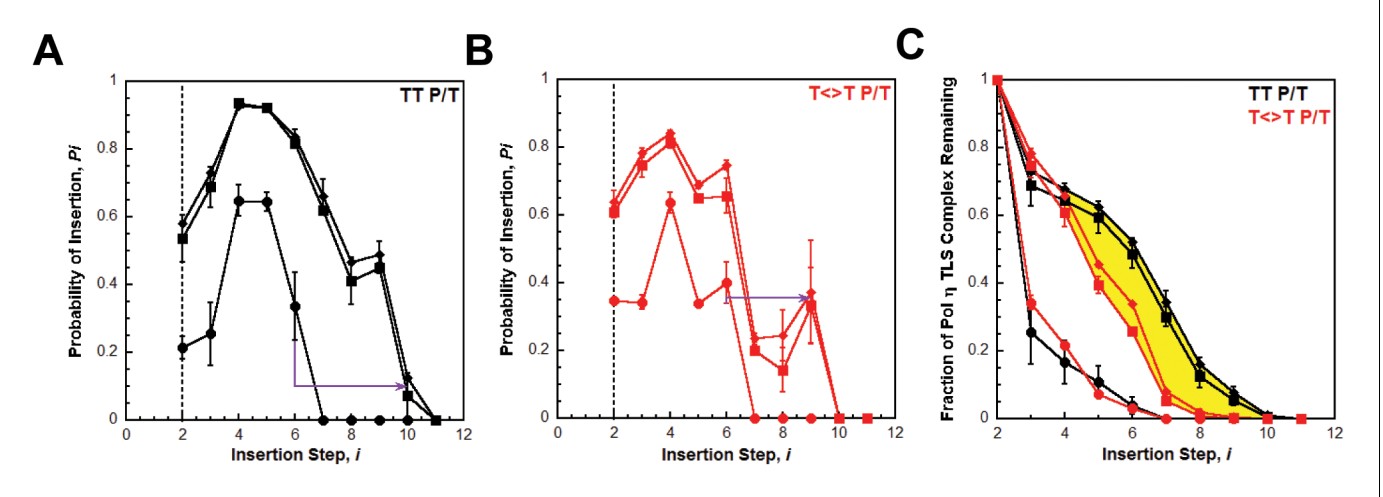

**Figure 5.** Processive DNA synthesis by pol η. Probability of insertion ($P_i$) for each dNTP insertion step ($i$) beyond the first was calculated for the experiments depicted in *Figure 4*. Results obtained in the absence of PCNA (•), in the presence of PCNA (■), and in the presence of (Ub)$_3$-PCNA (♦) are plotted versus the insertion step up to $i = 12$ and each data point represents the average ± SD of at least 3 independent experiments. dNTP insertion beyond $i = 10$ was not observed for pol η. (A –B) Processivity of pol η on undamaged (panel A, TT P/T) and damaged (panel B, T<>T P/T) P/T DNA substrates. The dashed line denotes the population of pol η that inserts a dNTP across from the 5′ T of a given template prior to dissociation, that is the pol η TLS complex. The arrows denote the increase in the extent of continuous DNA synthesis observed in the presence of a PCNA. (C) Dissociation of the pol η TLS complex. The fraction of the Pol η TLS complex (y) remaining at each insertion step, $i$, was determined for the undamaged (TT P/T) and damaged (T<>T P/T) P/T DNA substrates using the $P_i$ values reported in panels A and B. Results obtained in the absence of PCNA (•), in the presence of PCNA (■), and in the presence (Ub)$_3$-PCNA (♦) are plotted versus the insertion step, $i$. The divergence observed in the presence of a PCNA for the damaged and undamaged substrates is highlighted in yellow.

also responsible for the error-free replication of CPDs during TLS (*Yoon et al., 2009*). Next, we characterized the activity of pol η on a P/T DNA substrate containing a thymine-thymine CPD (TT-CPD) directly abutting the P/T junction (T<>T P/T, *Figure 4A*).

## TLS across a CPD by pol η is independent of PCNA monoubiquitination

As observed for the undamaged substrate, primer extension on the T<>T P/T (that is damaged) substrate is prominent with pol η alone (*Figure 4D*) and the $P_i$ values (*Figure 5B*) are low such that dNTP insertion beyond $i = 6$ is not observed. $P_2$ is increased 1.65 ± 0.221 fold on the damaged substrate, confirming that pol η is inherently more efficient at replicating a TT-CPD compared to a native sequence (*McCulloch et al., 2004*). However, the abundance of the pol η TLS complex is still limited ($P_2 = 0.349 ± 0.00298$), suggesting that DNA synthesis by pol η alone is inefficient for cellular TLS as it necessitates multiple binding encounters. Primer extension on the damaged substrate is stimulated to the same extent in the presence of PCNA and (Ub)$_3$-PCNA (*Figure 4D*), indicating that the binding of pol η to a PCNA encircling a damaged P/T junction is independent of PCNA monoubiquitination. Furthermore, the observed stimulations are approximately the same as those observed on the undamaged DNA substrate, indicating that a TT-CPD has no effect on the loading of PCNA onto P/T DNA, formation of a pol η holoenzyme, or the ensuing insertion of a dNTP across from the 3′T. In the presence of a PCNA, $P_i$ values up to $i = 6$ (*Figure 5B*) for the damaged substrate are increased compared to pol η alone and the observed stimulations are equivalent for PCNA and (Ub)$_3$-PCNA. Insertion of dNTPs beyond $i = 6$ is only observed in the presence of a PCNA on the damaged substrate and the measured $P_i$ values are equivalent in the presence of PCNA and (Ub)$_3$-PCNA. These behaviors agree with that observed for the undamaged substrate and indicate that the ubiquitin moieties conjugated to PCNA have no effect on DNA synthesis by a pol η holoenzyme.

Interestingly, the abundance of the pol η TLS complex ($P_2$) in the presence of a PCNA is approximately the same for the undamaged (*Figure 5A*) and damaged (*Figure 5B*) substrates. Thus, a pol η holoenzyme replicates a TT-CPD lesion and a native TT sequence with the same efficiency. Furthermore, $P_3$ values measured in the presence of a PCNA are approximately the same for each

substrate, indicating that a pol η holoenzyme has the same probability of extending a primer one dNTP beyond the 5′ T of a TT-CPD lesion as it does for the native TT sequence. Thus, binding of pol η to a PCNA encircling a P/T junction generically increases the efficiency of DNA synthesis across an adjacent di-pyrimidine sequence. However, a clear divergence in the behavior of pol η on the damaged and undamaged substrates is evident in the presence of a PCNA. $P_i$ for both substrates peaks at $i = 4$ and then decreases. On the undamaged substrate (*Figure 5A*), $P_4$ is higher and $P_i$ decreases such that >12.5% of the pol η TLS complex extends the primer an additional 6 dNTPs (*Figure 5C*). Insertion of dNTPs is not observed beyond $i = 10$. On the damaged substrate (*Figure 5B*), $P_4$ is lower and the descent of $P_i$ is much more rapid such that <2% of the pol η TLS complex extends the primer 6 dNTPs beyond the TT-CPD and dNTP insertion is not observed beyond $i = 9$ (*Figure 5C*). This disparity is not observed in the absence of a PCNA. Thus, a pol η holoenzyme is far less efficient at continuously extending a primer beyond a CPD lesion. In other words, replication of a CPD by a pol η holoenzyme selectively promotes dissociation of pol η downstream of the di-pyrimidine sequence. Perhaps the abnormally structured base pairs downstream of a CPD (*Park et al., 2002*) weaken the affinity of pol η for the P/T junction (that is increase $k_{off}$) and/or slow down dNTP insertion (i.e., decrease $k_{pol}$) (*Kusumoto et al., 2004*).

It should be noted that the $P_i$ values measured for pol η in the presence of a PCNA (*Figures 4* and *5*) account for dNTP insertions catalyzed by pol η alone and pol η holoenzymes up to and including $i = 6$. Based on the binding affinities of pol η for P/T DNA and a PCNA measured in this study (*Table 1*), only a fraction (~60%) of the pol η bound to the P/T DNA is also bound to the PCNA encircling the P/T junction under the conditions of the assay described in *Figure 4* Thus, each dNTP insertion step $\leq i = 6$ is carried out by a combination of pol η alone and pol η holoenzymes and, hence, the measured $P_i$ values account for both events. However, by directly comparing the activity of pol η in the absence and presence of a PCNA on both damaged and undamaged substrates, it is clearly evident that the majority of dNTP insertions $\leq i = 6$ are catalyzed by pol η holoenzymes when a PCNA is present. In particular, the rapid dissociation of the pol η TLS complex in the absence of a PCNA (*Figure 5C*) indicates that dNTP insertion beyond $i = 2$ is predominantly catalyzed by pol η holoenzymes when a PCNA is present. Thus, $P_i$ values measured in the presence of a PCNA for $i < 6$ are likely an underestimate for pol η holoenzymes due to the contribution of pol η alone. Insertion of dNTPs by pol η beyond $i = 6$ is only observed in the presence of a PCNA and, thus, only reflects the activity of pol η holoenzymes.

## Discussion

In this report, we provide extensive quantitative evidence that the binding of pol η to PCNA and DNA synthesis by a pol η holoenzyme are both independent of PCNA monoubiquitination. Thus, direct binding of pol η to the ubiquitin moieties conjugated to PCNA, if it occurs, is dispensable for pol η-mediated TLS across a UV-induced CPD lesion, in agreement with in vivo studies. This refutes the recruitment/displacement model for human TLS and, hence, indicates that pol switching occurs independently of PCNA monoubiquitination. In support of our conclusion, we compared the activities of pols δ and η in various experimental contexts. Together with previous reports from our lab, the results from these studies indicate that pol switching during TLS on the lagging strand is an efficient and passive process that occurs independently of PCNA monoubiquitination, as discussed below (*Figure 6*).

### Polymerase exchange during TLS

Pol δ, a replicative B-family pol, inserts complementary dNTPs very fast ($k_{pol} \sim 100\ s^{-1}$) and maintains very high fidelity (*Hedglin et al., 2016*; *Schmitt et al., 2009*). However, pol δ alone has very low affinity for P/T DNA and must anchor to PCNA to efficiently replicate DNA (*Hedglin et al., 2016*). Pol δ binds much tighter to PCNA encircling DNA than to DNA alone (*Table 1*) and effectively captures PCNA encircling a P/T junction to initiate DNA synthesis (*Hedglin et al., 2013b*). A pol δ holoenzyme inserts dNTPs much faster (>700 fold) than pol δ dissociates from PCNA encircling DNA (*Hedglin et al., 2016*). Thus, anchoring to PCNA dramatically increases the $P_i$ of pol δ from 0.0 to ~1.0 such that the extent of continuous dNTP insertion (that is processivity) increases by kilobases (*Figure 4—figure supplement 1*). Upon encountering a lesion it cannot accommodate, such as a UV-induced CPD, pol δ rapidly and passively dissociates into solution, leaving PCNA behind on the

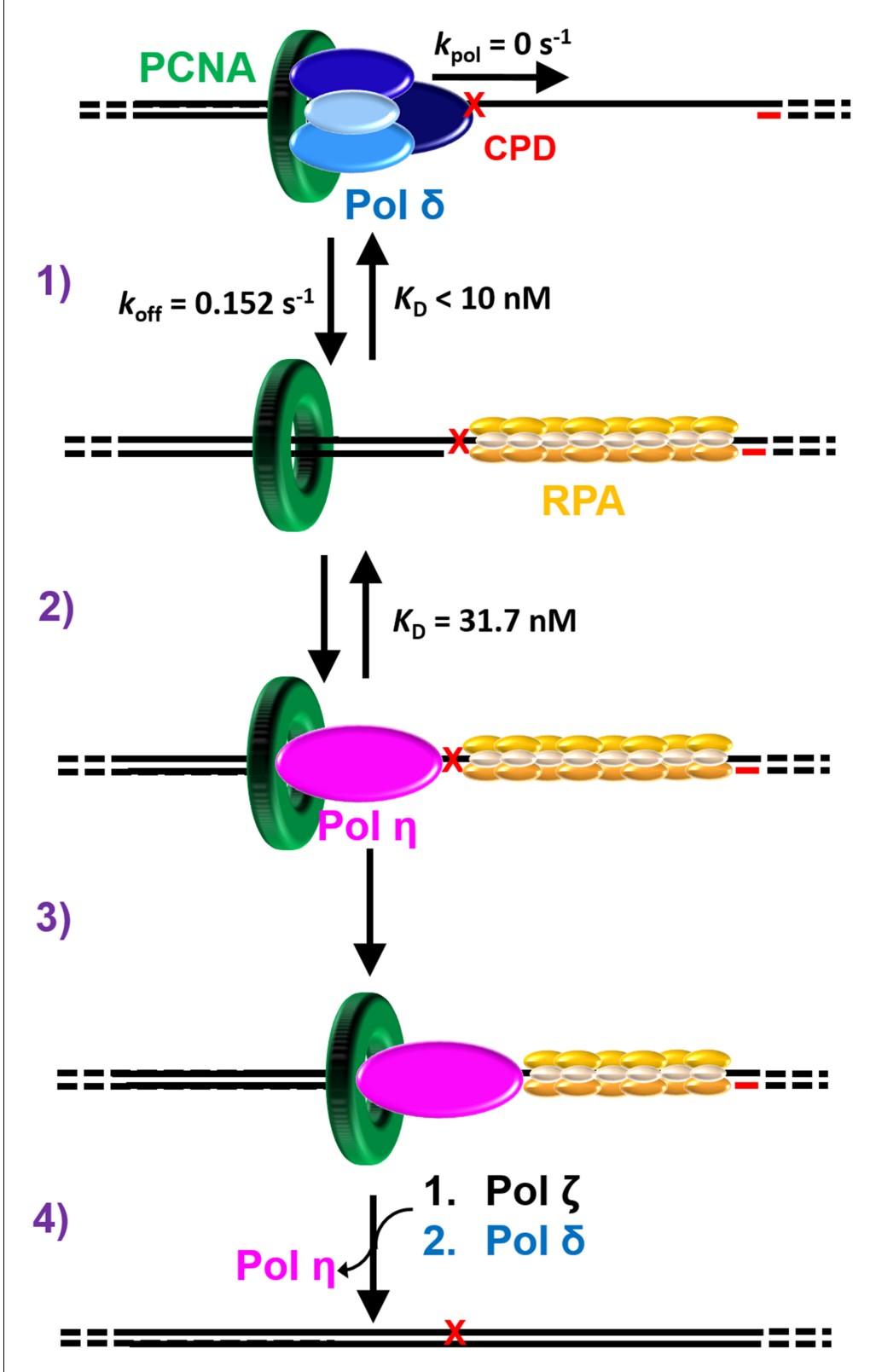

**Figure 6.** Polymerase exchange during TLS on the lagging strand. While replicating a lagging strand template, pol δ maintains a loose association with PCNA while inserting dNTPs very fast. (1) Upon encountering a DNA lesion (X) it cannot accommodate, such as UV-induced CPD, pol δ rapidly dissociates from DNA, leaving PCNA behind. Pol δ may re-bind to the resident PCNA but pol δ-mediated DNA synthesis cannot resume until the lesion is replicated by one or more TLS pols. (2) The binding affinity of pol η for P/T DNA drives the passive exchange of pols at a blocked P/T junction. (3)
*Figure 6 continued on next page*

*Figure 6 continued*

Once localized to the blocked P/T junction, the resident PCNA stabilizes pol η on the DNA, increasing the efficiency of the ensuing TLS. (**4**) After replicating the CPD, dissociation of pol η is promoted downstream of the damage. This provides an efficient and generic pathway for pol η to depart a damaged template and, hence, limits the extent of error-prone DNA synthesis by pol η downstream of a CPD. In the event that pol η-mediated TLS across and beyond a CPD lesion terminates with a mismatched P/T junction, another TLS pol, such as pol ζ, may access the P/T junction upon pol η's departure and faithfully extend the mismatch, allowing DNA synthesis by pol δ to resume.

DNA (*Hedglin et al., 2016*). Pol δ may re-bind to the PCNA residing at a blocked P/T junction but pol δ-mediated DNA synthesis cannot resume until the lesion is replicated (*Figure 6*, step 1). This suggests that pol η may access the blocked P/T junction in the interim that pol δ has vacated. Our previous study demonstrated that this pol exchange is independent of PCNA monoubiquitination and rate-limited by the rapid and passive dissociation of pol δ from the resident PCNA (*Hedglin et al., 2016*). The current study revealed that pol η binds much tighter to P/T DNA than to a PCNA (*Table 1*) such that pol exchange is driven by the binding affinity of pol η for the P/T DNA (*Figure 3*). Thus, pol exchange during TLS on the lagging strand is a competition between pol η binding to a blocked P/T junction and pol δ binding to PCNA encircling a blocked P/T junction (*Figure 6*, step 2).

The binding affinity of pol η for PCNA is substantial and independent of PCNA monoubiquitination (*Table 1*). Once localized to a P/T junction, pol η binds to the resident PCNA, forming a holoenzyme. This association stabilizes pol η at a P/T junction and increases the efficiency of the ensuing DNA synthesis across a generic di-pyrimidine sequence directly abutting the P/T junction (*Figure 6*, step 3). However, a pol η holoenzyme is far less efficient at continuously extending a primer beyond a CPD compared to the native sequence (*Figures 4* and *5*). Thus, replication of a CPD by a pol η holoenzyme selectively promotes dissociation of pol η downstream of the di-pyrimidine sequence. This provides an efficient and generic pathway for pol η to depart a damaged template and, hence, limits the extent of error-prone DNA synthesis by pol η downstream of a CPD. This may account for the remarkably low error-rates observed in human cells for pol η-mediated TLS after UV irradiation (*Yoon et al., 2009*). Pol δ is inefficient at extending mismatched primers (*Meng et al., 2010*). In the event that pol η-mediated TLS across and beyond a CPD lesion terminates with a mismatched P/T junction, another TLS pol, such as pol ζ, may access the P/T junction upon pol η's departure and faithfully extend the mismatch (*Makarova and Burgers, 2015*), allowing DNA synthesis by pol δ to resume (*Figure 6*, step 4).

## Role of monoubiquitinated PCNA during TLS

Monoubiquitination of PCNA is imperative for replication through CPD lesions in human cells (*Yoon et al., 2012*) and the signal for this PTM is the build-up and persistence of RPA-coated ssDNA downstream of blocked P/T junctions (*Hedglin and Benkovic, 2015*). In human cells irradiated with UV at the $G_1$/S border, RPA foci rapidly built up, persisted through S-phase, and then disappeared primarily in $G_2$/M phase when the bulk of chromosomal replication had been completed (*Diamant et al., 2012*). An earlier study observed a similar trajectory for monoubiquitinated PCNA (*Niimi et al., 2008*), which only resides on DNA in vivo (*Hedglin and Benkovic, 2015*). Thus, the role (s) of monoubiquitinated PCNA during TLS may be initiated at the onset of S-phase and persist into $G_2$/M. Given the large size of ssDNA stretches (100's to 1000's of nucleotides) and the remarkably low error rates observed in vivo after UV exposure, this argues strongly against monoubiquitinated PCNA selectively and directly promoting error-prone DNA synthesis by TLS pols (*Hedglin and Benkovic, 2015*; *Lange et al., 2011*; *Yoon et al., 2012*). This study demonstrates that the binding of pol η to PCNA and the ensuing TLS across a UV-induced CPD are both independent of PCNA monoubiquitination. Furthermore, pol ζ, a B-family TLS pol implicated in TLS following UV exposure (see above), does not contain a ubiquitin-binding domain yet assembly of pol ζ into replication foci at UV-induced lesions requires monoubiquitination of PCNA by Rad6/Rad18 (*Makarova and Burgers, 2015*; *Yoon et al., 2015*). Together, this suggests that the role(s) of PCNA monoubiquitination in human TLS across and past a UV-induced DNA lesion is indirect. A recent study from our lab demonstrated that assembly/disassembly of a pol δ holoenzyme and continuous DNA synthesis by a pol δ holoenzyme are both independent of PCNA monoubiquitination (*Hedglin et al., 2016*). The

ubiquitin moieties attached to PCNA do not alter the conformation of the sliding clamp ring (*Dieckman et al., 2012*) nor do they shelter the region of PCNA that interacts with DNA pols (*Tsutakawa et al., 2011*, *2015*). Hence, monoubiquitination of PCNA is not envisioned to have any effect on DNA synthesis by a pol ε holoenzyme on a leading strand template. Collectively, this argues against PCNA monoubiquitination indirectly promoting TLS by destabilizing the replicative pol holoenzymes. In addition, this suggests the ubiquitin moieties attached to PCNA do not need to be removed immediately after TLS so that DNA synthesis by the replicative pols can resume. Indeed, de-ubiquitination of PCNA in human cells irradiated with UV does not occur until after the bulk of DNA synthesis (that is S-phase) has been completed (*Niimi et al., 2008*). We propose that the ubiquitin moieties conjugated to PCNA indirectly promote DNA synthesis in general at sites of DNA damage. One possibility is that monoubiquitinated PCNA alters the local chromatin structure encompassing a DNA lesion to promote DNA synthesis past and beyond the offending damage. In a recent study on human cells, pol η was transiently immobilized within replication foci after treatment with DRAQ5, a DNA intercalating agent that temporarily disrupts chromatin structure without eliciting PCNA monoubiquitination or a DNA damage response (*Sabbioneda et al., 2008*). This suggests that transient opening of chromatin promotes access of pol η to DNA. In the same study, accumulation of pol η into replication foci following UV irradiation was independent of PCNA monoubiquitination but monoubiquitinated PCNA increased the residence time of pol η within the foci, similar to DRAQ5 treatment. Thus, it is possible that the ubiquitin moieties conjugated to PCNA facilitate the exposure of DNA at blocked P/T junctions. In humans, chromatin assembly factor 1 (CAF-1) deposits histones H3/H4 on nascent DNA immediately after passage of the replication fork, initiating nucleosome assembly. This process is mediated by a direct interaction between CAF-1 and PCNA left behind on the DNA (*Shibahara and Stillman, 1999*). In cell-free extracts, CAF-1 efficiently triggers a progressing wave of nucleosome assembly on a gapped DNA substrate in the absence of DNA synthesis (*Moggs et al., 2000*). Such activity may impede TLS in vivo by limiting the access of DNA pols to sites of DNA damage. The amino acid residues within a PCNA ring that are required for CAF-1 mediated histone deposition are far removed from the region of PCNA that interacts with replication proteins, including pols δ and η (*Zhang et al., 2000*). Results from computational studies suggest that the ubiquitin moieties conjugated to PCNA at lysine residue(s) K164 could selectively block the CAF-1 binding sites on PCNA (*Tsutakawa et al., 2015*). This may preclude the binding of CAF-1 to PCNA encircling a blocked P/T junction, inhibiting histone deposition until the offending DNA lesion is replicated. Further studies will test this hypothesis.

## Materials and methods

### Oligonucleotides

Oligonucleotides were synthesized by Integrated DNA Technologies (Coralville, IA) and purified on denaturing polyacrylamide gels. Concentrations were determined from the absorbance at 260 nm using the calculated extinction coefficients. The P/T DNA substrates (*Figure 2—figure supplement 1*) were annealed as previously described (*Hedglin et al., 2016*). For experiments in which DNA synthesis was measured, the primer was 5′-labeled with $^{32}$P as previously described (*Hedglin et al., 2016*).

### Recombinant human proteins

Catalytically-inactive ('dead') pol η was prepared by mutating residues D115 and E116 within the catalytic core of human pol η to alanine (D115A/E116A) using forward (5′-GAACGTGCCAGC A TTGCTGCGGCTTACGTAGATCTG-3′) and reverse (5′-CAGATCTACGTAAGCCGCAGCAATGC TGGCA CGTTC-3′) primers to the pET21A plasmid expressing the full-length, human pol η protein. These amino acid residues are necessary for catalytic activity. In vivo, this full-length, human pol η mutant retains all biological functions except DNA synthesis activity (*Bergoglio et al., 2013*; *Durando et al., 2013*; *Rey et al., 2009*). Wild-type and catalytically-inactive pol η was purified from the pET21A-hPol η expression vector by slight modifications of a published protocol (*Li et al., 2013*). The protein concentration was determined from the calculated extinction coefficient (64010 $M^{-1}cm^{-1}$) and verified by Bradford assay using BSA as a protein standard. For N-terminally labeling pol η with Cy3 NHS ester (GE healthcare UK Limited, Buckinghamshire, UK), Pol η was exchanged

into labeling buffer (20 mM HEPES pH 6.5, 200 mM NaCl, 10% glycerol), mixed with a five-fold excess of Cy5 NHS ester, incubated for 1 hr at 4°C, and then loaded onto a G25 size exclusion column (GE healthcare, Uppsala, Sweden) that had been pre-equilibrated with Pol η storage buffer (25 mM tris pH 7.5, 250 mM NaCl, 10% glycerol and 5 mM β-Mercaptoethanol). The column was developed with Pol η storage buffer and the labeled protein fraction was collected and concentrated. The labeling efficiency (1.6 Cy3 dyes/pol η protein on average) was calculated according to the manufacturer's protocol. The concentration of active pol η was determined by active site titration according to a published protocol (*Washington et al., 2003*). Wild-type and Cy3-labeled retained more than 95 and 80% activity, respectively. All concentrations of pol η indicated throughout the text refer to the final concentration of active protein. Exonuclease-deficient pol δ was prepared by mutating D402 in the catalytic (POLD1) subunit to alanine (D402A) using forward (5'-CAACATCCAGAACTTC GCCCTTC CGTACCTCATC-3') and reverse (5'-GATGAGGTACGGAAGGGCGAAGTTCTGGATG TTG-3') primers to the plasmid expressing the wild-type POLD1 subunit (*Masuda et al., 2007*). This pol δ mutant was characterized previously (*Hedglin et al., 2016*), used in all of the primer extension assays reported herein, and is referred to simply as pol δ throughout the text. Wild-type PCNA (PCNA), Cy5-labeled PCNA (Cy5-PCNA), monoubiquitinated PCNA ((Ub)$_3$-PCNA), exonuclease-deficient pol δ (Pol δ), RPA, and a truncated form of RFC (RFC) were obtained as described in a previous report (*Hedglin et al., 2016*).

All experimental procedures described below were performed at room temperature in 1X replication assay buffer (25 mM TrisOAc, pH 7.7, 10 mM Mg(OAc)$_2$, 125 mM KOAc) supplemented with 0.1 mg/ml BSA, 1 mM DTT and the ionic strength was adjusted to physiological (200 mM) with KOAc.

## Steady state FRET measurements

FRET measurements were done in a Jobin Yvon fluoromax-4 fluorimeter. The fluorescence emission intensity maxima for the Cy3 donor ($I_{570}$) and the Cy5 acceptor ($I_{670}$) occur at 570 nm and 670 nm, respectively. In the presence of both Cy3-pol η and Cy5-PCNA, a FRET is indicated by an increase in $I_{670}$ and a concomitant decrease in $I_{570}$. This can be quantified by monitoring the $I_{670}/I_{570}$ ratio. For the experiments depicted in *Figure 1—figure supplement 2*, Cy5-PCNA (20 nM) and Cy3-Pol η (200 nM) were pre-incubated, excited at 514 nm, and the fluorescence emission intensity was recorded from 540 to 750 nm. For the direct titration experiments depicted in *Figure 1A* and 20 nM Cy3-Pol η was pre-incubated with increasing concentrations of Cy5-PCNA, excited at 514 nm, and the fluorescence emission intensity at 570 nm (Cy3 donor fluorescence emission maximum, $I_{570}$) was measured. In order to account for the signal generated from Cy5-PCNA, each value was corrected for by the corresponding $I_{570}$ value obtained in the absence of pol η. The donor fluorescence quenching efficiency ($E_{dq}$) was then calculated at each Cy5-PCNA concentration by the equation $E_{dq} = 1-I_{DA}/I_D$, where $I_{DA}$ and $I_A$ are the $I_{570}$ values obtained in the presence and absence of Cy5-PCNA, respectively. The data was analyzed as described below for an equilibrium binding assay. For the competitive titration experiments depicted in *Figure 1C*, the Cy3-Pol η•Cy5-PCNA complex (200 nM Cy3-Pol η, 20 nM Cy5-PCNA) was pre-assembled, an unlabeled PCNA was added (either PCNA or (Ub)$_3$-PCNA, 0–5 μM), and then the $I_{670}/I_{670}$ ratio was measured. As a control to demonstrate that Cy3 does not affect the interaction between pol η and PCNA, these assays were repeated with pol η (0–5 μM) as the unlabeled competitor (*Figure 1—figure supplement 3*). The data was analyzed as described below for competitive inhibition assays. For holoenzyme assembly depicted in *Figure 3—figure supplement 1*, RFC (100 nM), Cy5-PCNA (100 nM) and Cy3-P/T DNA (100 nM) were pre-incubated in the presence of 1 mM ATP. Increasing concentrations of unlabeled Pol η were then added and the $I_{670}/I_{570}$ ratio was measured. The data was analyzed as described below for equilibrium binding assays. For the FRET-based competition assays (*Figure 3—figure supplement 2*), the competitive primer extension assays (*Figure 3*) were repeated using a TT P/T DNA substrate containing a 5' Cy3-label on the primer (Cy3-TT P/T, *Figure 2—figure supplement 1*) and Cy5-PCNA. Under these conditions, pol δ (50 nM) stabilizes Cy5-PCNA (50 nM) on the Cy3-TT P/T DNA substrate (10 nM), maximizing the FRET signal. After a 5 min incubation with various concentrations of 'dead' pol η (0–1 μM), the FRET signal ($I_{670}/I_{570}$) was measured. As a control, FRET was measured in the absence of pol δ and pol η.

## Fluorescence anisotropy

10 nM of the fluorescein-labeled P/T DNA substrate (FLUOR-P/T DNA, *Figure 2—figure supplement 1*) was titrated with various concentrations of either pol η or pol δ and the fluorescence polarization was measured.

## Primer extension assays

Single-turnover primer extension assays were carried out with pols δ or η exactly as described in a previous report (*Hedglin et al., 2016*). Erroneous insertion of ATP during the pre-incubation period did not occur with pol δ under any experimental condition. For pol η, such events did not occur beyond $i = 2$ and were only notable (that is >10% of the total amount of primer extension products) for the T<>T P/T DNA substrate in the absence of a PCNA. Controls experiments were carried out to account for such activity. The primer extension assays described in *Figure 4* were repeated except aliquots were quenched prior to the addition of the trap/dNTPs. For each condition, the concentration of each primer extension product observed during the pre-incubation was subtracted from the corresponding primer extension products observed over the time course of the experiment. The probability of insertion ($P_i$) was calculated as described in a previous report (*Hedglin et al., 2016*). The pre-assembled pol η•P/T DNA complex that replicates the first two nucleotides of a given template within a single binding encounter is $P_2$ and is referred to as the 'Pol η TLS complex.' The fraction of the Pol η TLS complex (y) remaining at each insertion step, $i$, was determined by normalizing $P_2$ to 1.0 and multiplying by the $P_i$ values for each involved step. For example, the fraction of the pol η TLS complex remaining after 2 dNTP insertion is $y = 1.0 \times P_3 \times P_4$.

For the competition assays, the pol δ holoenzyme was pre-assembled as in the single-turnover primer extension assays (*Hedglin et al., 2016*) with minor differences in the final concentrations: 10 nM TT P/T DNA, 40 nM Neutravidin, 50 nM RPA, 50 nM PCNA trimer (either wild type or (Ub)$_3$-PCNA), 1 mM ATP, 10 nM RFC, and 50 nM pol δ. The pol δ holoenzyme was pre-incubated for 1 min prior to the addition of catalytically-inactive ('dead') pol η (0–5 µM). After 5 min, DNA synthesis by surviving pol δ holoenzymes was initiated by addition of dNTPs (100 µM of each). After 30 s, the reactions were quenched and analyzed by denaturing PAGE. For each time point, the fraction of pol δ holoenzyme remaining was calculated by dividing the concentration of full-length product (62-mer) by the total concentration of primer extension products.

## Data analysis

For all equilibrium binding experiments, the raw data (y) was fit to a one-site binding model (*Equation 1*) where $\mathrm{R}$ is the range, $K_D$ is the equilibrium binding constant, and $\mathrm{C}$ is a constant.

$$y = \frac{\mathrm{R}[\mathrm{Ligand}]_{\mathrm{Total}}}{K_D + [\mathrm{Ligand}]_{\mathrm{Total}}} + \mathrm{C} \tag{1}$$

In all experiments, the initial concentration of the substrate is substantial and, hence, depletes the concentration of added ligand ([Ligand]$_{\mathrm{Total}}$), that is [Ligand]$_{\mathrm{free}}$<[Ligand]$_{\mathrm{Total}}$. In order to obtain the most accurate value for $K_D$, each experiment was normalized by utilizing the values for and to calculate the fractional saturation of the substrate and [Ligand]$_{\mathrm{Free}}$. The fractional saturation ($\mathrm{F}$) of the substrate was plotted versus [Ligand]$_{\mathrm{Free}}$ and fit to a one-site binding model (*Equation 2*) where $\mathrm{Y}_{\max}$ is the maximum specific binding and is equal to 1.0 (that is, the range).

$$\mathrm{F} = \frac{\mathrm{Y}_{\max}[\mathrm{Ligand}]_{\mathrm{Free}}}{K_D + [\mathrm{Ligand}]_{\mathrm{Free}}} \tag{2}$$

For all competitive inhibition experiments, the measured experimental signal (y) was plotted versus the log concentration of the respective competitor and the data was fit to a dose-response inhibition model (*Equation 3*) where $\mathrm{Y}_{\min}$ and $\mathrm{Y}_{\max}$ are the minimum and maximum experimental signals, respectively, and $\mathrm{IC}_{50}$ is the concentration of competitor that gives a response halfway between $\mathrm{Y}_{\max}$ and $\mathrm{Y}_{\min}$.

$$y = \mathrm{Y}_{\min} + \left[ \frac{\mathrm{Y}_{\max} - \mathrm{Y}_{\min}}{1 + 10^{([\mathrm{Competitor}] - \mathrm{LogIC}_{50})}} \right] \tag{3}$$

Each data set was then normalized using the respective $Y_{min}$ and $Y_{max}$ values to plot the fraction $(F)$ of the pre-assembled complex remaining versus the log concentration of the respective competitor and fit to a dose-response inhibition model (*Equation 4*) where $Y_{max}$ is the range and equal to 1.0.

$$y = \left[ \frac{Y_{max}}{1 + 10^{([\text{Competitor}] - \text{LogIC}_{50})}} \right] \qquad (4)$$

Under the conditions of the competitive inhibition assays, the Cheng-Prusoff correction for obtaining the dissociation constant for the competitive inhibitor (that is, inhibition constant, $K_i$) from the experimentally measured $IC_{50}$ and values is not exact. We utilized the exact solution (*Equation 5*) provided by Munson and Rodbard (*Munson and Rodbard, 1988*; *Nikolovska-Coleska et al., 2004*) where $L_T$ is the total concentration of the ligand that the competitive inhibitor will bind to and $y_0$ is the initial bound to free ratio for this ligand prior to the addition of the competitive inhibitor.

$$K_i = \frac{IC_{50}}{1 + \frac{L_T(y_0+2)}{2K_D(y_0+1)} + y_0} - K_D\left(\frac{y_0}{y_0+2}\right) \qquad (5)$$

In *Figure 1C*, $L_T$ and $y_0$ refer to Cy3-pol η and $y_0$ is calculated from the measured $K_D$ for the Cy3-pol η•Cy5-PCNA complex (*Figure 1B*). In *Figure 1—figure supplement 3*, $L_T$ and $y_0$ refer to Cy5-PCNA and $y_0$ is calculated from the measured $K_D$ for the Cy3-pol η•Cy5-PCNA complex (*Figure 1B*). In *Figure 3*, $L_T$ and $y_0$ refer to the TT P/T DNA•PCNA complex and $y_0$ is calculated from the $K_D$ reported for the pol δ•PCNA•P/T DNA complex (*Zhou et al., 2012*).

## Acknowledgements

This work was supported by NIH Grant GM13306 (SJB). We would like to acknowledge Dr. Anthony Berdis (Cleveland State University, Cleveland, OH) and Dr. John B Hays (Oregon State University) who graciously provided the oligonucleotide containing a TT-CPD and the plasmid expressing full-length human pol η, respectively.

## Additional information

### Funding

| Funder | Grant reference number | Author |
| --- | --- | --- |
| National Institutes of Health | GM13306 | Stephen J Benkovic |

The funders had no role in study design, data collection and interpretation, or the decision to submit the work for publication.

### Author contributions

MH, Conception and design, Acquisition of data, Analysis and interpretation of data, Drafting or revising the article; BP, Conception and design, Acquisition of data, Drafting or revising the article; SJB, Conception and design, Analysis and interpretation of data, Drafting or revising the article

### Author ORCIDs

Stephen J Benkovic, http://orcid.org/0000-0003-3680-3481

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
