## [Decision Letter]

Thank you for submitting your article "Characterization of human translesion DNA synthesis across a UV-induced DNA lesion" for consideration by *eLife*. Your article has been favorably evaluated by John Kuriyan as the Senior Editor and three reviewers, one of whom, Michael R Botchan (Reviewer #1), is a member of our Board of Reviewing Editors.

The reviewers have discussed the reviews with one another and the Reviewing Editor has drafted this decision to help you prepare a revised submission.

Summary:

This paper describes mostly equilibrium experiments to address the question whether PCNA ubiquitylation has a direct role in TLS at lesions by promoting recruitment or activity of the TLS polymerase η. The very beautiful biochemistry reported here makes it clear that the Ub modification of PCNA can't play a key role in recruitment for either polymerase η or δ given what we know about the pathway.

The experiments show clearly that:

1) The affinity of pol η for PCNA in solution in the absence of DNA or other factors is independent of the state of ubiquitination of PCNA.

2) The affinity of pol η for a primer terminus in the absence of other factors is high (4X affinity for free PCNA).

3) In an exchange experiment pol η can equilibrate with a primer terminus preloaded with PCNA-pol δ during five-minute incubation. PCNA remains associated with the DNA during the exchange of η for δ.

4) PCNA increases the processivity of pol η somewhat, but in the presence of PCNA, RF-C, and RPA, synthesis across a CPD by pol η is largely independent of the ubiquitination status of PCNA. Copying a CPD is associated with a higher rate of dissociation downstream.

Essential revisions:

1) The Discussion (and probably the Introduction) should be rewritten to focus on issues that are directly addressed by the experiments presented. In particular, much of the Discussion is devoted to analyzing two models for the timing of TLS – the "on-the-fly" model in which an exchange to a TLS pol occurs at the fork and the post-replicative gap model in which TLS occurs in gaps left behind after replication restarts beyond the lesion. Distinguishing between these two models requires kinetic arguments and the authors make such arguments based on data in other papers. But this paper has no kinetics in it and really doesn't contribute much to the distinction between the models, much less to the grand synthesis model presented. The observation that the binding and activity of pol η are not affected by the state of ubiquitination of PCNA could be easily accommodated by either model. It would be appropriate in our view to stick to the main conclusions that can be drawn from the data and their implications – these relate to the question of whether PCNA ubiquitination plays a direct role in TLS. The authors state their conclusion at the beginning of the discussion, but then spend most of the time speculating about issues that are not germane to the experiments. A concise speculation about what role Ub-PCNA plays in the TLS pathway is relevant since the genetics show that this is required at some step.

The study focuses on Pol η and Pol δ. It might be informative for the authors to comment briefly in the Discussion on whether/or how the work might relate to a possible function of Ubi-PCNA for synthesis by other TLS Pols and also the leading strand polymerase – or should we take home that Ubi-PCNA has zero role on either strand?

2) Experimentally the issue we would like to have addressed with a figure is that the authors add to the paper something they have likely done and possibly shown as a figure in their earlier work. Considering that various results compare PCNA and Ubi-PCNA and are negative (i.e. no difference between wtPCNA and Ubi-PCNA), there will be many readers that would appreciate seeing an SDS PAGE of the wtPCNA and the Ubi-PCNA, proving the PCNA is fully ubiquinated.

---

## [Author Response]

*[…] Essential revisions:*

*1) The Discussion (and probably the Introduction) should be rewritten to focus on issues that are directly addressed by the experiments presented. In particular, much of the Discussion is devoted to analyzing two models for the timing of TLS – the "on-the-fly" model in which an exchange to a TLS pol occurs at the fork and the post-replicative gap model in which TLS occurs in gaps left behind after replication restarts beyond the lesion. Distinguishing between these two models requires kinetic arguments and the authors make such arguments based on data in other papers. But this paper has no kinetics in it and really doesn't contribute much to the distinction between the models, much less to the grand synthesis model presented. The observation that the binding and activity of pol η are not affected by the state of ubiquitination of PCNA could be easily accommodated by either model. It would be appropriate in our view to stick to the main conclusions that can be drawn from the data and their implications – these relate to the question of whether PCNA ubiquitination plays a direct role in TLS. The authors state their conclusion at the beginning of the discussion, but then spend most of the time speculating about issues that are not germane to the experiments. A concise speculation about what role Ub-PCNA plays in the TLS pathway is relevant since the genetics show that this is required at some step.*

*The study focuses on Pol η and Pol δ. It might be informative for the authors to comment briefly in the Discussion on whether/or how the work might relate to a possible function of Ubi-PCNA for synthesis by other TLS Pols and also the leading strand polymerase – or should we take home that Ubi-PCNA has zero role on either strand?*

The Abstract, Introduction, and Discussion have been re-written to focus on the issue that is directly addressed by the experiments presented; polymerase switching during human TLS and the role of PCNA monoubiquitination in this process. Specifically;

The Introduction has been revised to focus on the issue to be addressed; the role of monoubiquitinated PCNA in polymerase switching during human TLS.

The Discussion has been revised to focus on the main conclusions that can be drawn from the data and their implications; polymerase exchange during human TLS is independent of PCNA monoubiquitination and PCNA monoubiquitination has an indirect role in TLS.

All discussion on the two models for the timing of TLS and the distinction between them has been removed (from the Abstract, main text, and figure legends) and a concise speculation about the indirect role of monoubiquitinated PCNA in human TLS has been provided. Accordingly, Figure 6 has been revised to focus only polymerase switching during human TLS on the lagging strand.

Also, we have commented briefly in the Discussion about the function of monoubiquitinated PCNA for DNA synthesis by other TLS and the leading strand polymerase, pol ε, in response to UV radiation.

*2) Experimentally the issue we would like to have addressed with a figure is that the authors add to the paper something they have likely done and possibly shown as a figure in their earlier work. Considering that various results compare PCNA and Ubi-PCNA and are negative (i.e. no difference between wtPCNA and Ubi-PCNA), there will be many readers that would appreciate seeing an SDS PAGE of the wtPCNA and the Ubi-PCNA, proving the PCNA is fully ubiquinated.*

An SDS PAGE with wild-type PCNA and monoubiquitinated PCNA has been provided as a figure (Figure 1—figure supplement 4). This is indicated within the main text and the figure legend for Figure 1.